# SPLIT, NOT SPILLED: PRACTICAL OBFUSCATION-BASED PRIVACY-PRESERVING SPLIT LEARNING

## ABSTRACT

Split Learning (SL) partitions a deep neural network between client and server, enabling collaborative training while reducing the client's computational load. However, it has been shown that the intermediate activations ("smashed data") of the client's model, shared with the server, leak sensitive information. Existing defenses are limited: many assume only passive adversaries, degrade accuracy significantly, or have already been bypassed by recent reconstruction attacks.

In this work, we propose SEAL, a client-side obfuscation framework for SL. By applying secret, client-specific periodic transforms, SEAL creates an exponentially large, unsearchable function space that prevents reconstruction of smashed data. We rigorously characterize the class of periodic functions that yield orthogonal, reversible, and numerically stable transforms, ensuring both security and utility preservation. Extensive experiments on image and text benchmarks show that SEAL withstands state-of-the-art reconstruction attacks while maintaining high accuracy.

## 1 INTRODUCTION

Large-scale deep neural networks (DNNs) have achieved state-of-the-art results across computer vision, natural language processing, and healthcare applications. However, deploying such models on edge or resource-constrained devices remains challenging due to their significant computational and memory demands. Split Learning (SL) has emerged as a promising framework to mitigate these challenges by partitioning a DNN between client and server: lightweight layers remain on the client while the server executes the remaining heavy layers. This allows clients to benefit from large models without sharing raw data.

While SL is motivated by privacy concerns, recent work has revealed that the intermediate activations of the model shared by the client with the server ("smashed data") can still leak sensitive information. In particular, a malicious or even honest-but-curious server can reconstruct client inputs or infer sensitive attributes through inference attacks. These vulnerabilities undermine SL's core privacy goal and limit its adoption in sensitive domains.

Existing defenses either (i) rely on statistical dependence minimization (Vepakomma et al., 2019), which can be bypassed by stronger attacks (Pasquini et al., 2021); or (ii) adopt differential privacy (Abadi et al., 2016), which introduces severe utility degradation; one might also (iii) apply fixed transforms such as DCT (Discrete Cosine Transform) (Ahmed et al., 1974), DST (Discrete Sine Transform) (Jain, 1976), and wavelets (Mallat, 1989), which an adversary can brute-force and exploit (as the basis periodic functions in these transforms are fixed and well-known). Advanced cryptographic techniques like homomorphic encryption or Multiparty Computation, provide strong leakage resistance but impose heavy overheads, limiting their practicality for high-dimensional activations or constrained clients (Knott et al., 2021; Gilad-Bachrach et al., 2016).

In this work, we propose SEAL, a novel obfuscation framework for SL to prevent information leakage to the server. The core idea is that each client applies a secret, dynamic transform based on periodic functions to obfuscate its smashed data before sending it to the server. Unlike public transforms, the client's basis function is **unknown** to the server or any other participant, making the transformation client-specific, i.e., each client will be able to select its own secret periodic function for this transformation. This forces an adversary to brute-force over a large, unsearchable function

space that prevents reconstruction of smashed data. The transform space is exponential because each client selects independent parameters for its periodic function, with every combination yielding a distinct valid transform. As both the sampling resolution of the function and the dimensionality of the smashed data increase, the number of possible transforms grows exponentially Edelman et al. (1998).

However, to design such frequency transformations, two properties are essential: (i) orthogonality, ensuring independent transformed representations and reversibility, and (ii) energy compaction, retaining critical information in a few coefficients (mainly low-frequency components). Also, not all functions can form valid transform matrices. Non-periodic functions are excluded, as they lack a period and are unable to satisfy the orthogonality conditions required for invertible transforms.

Furthermore, only a subset of periodic functions can construct transformation matrices that are reversible, energy-preserving, and numerically stable. These periodic functions must form an *orthogonal system with respect to integration over a period* (Szeg, 1939; Katznelson, 2004), and the sampling nodes must be chosen so that the associated evaluation matrix is *non-singular and well-conditioned* (e.g., Chebyshev nodes) (Gautschi, 2004) (see Sect. 2.5). Hence, in SEAL, we detail the steps to ensure that client-chosen secret functions are valid periodic functions and then the obtained transformation preserves the properties of orthogonality and energy compaction.

Our contributions are as follows:

- We introduce SEAL, a technically involved client-side obfuscation framework for SL that leverages secret, dynamically chosen periodic transforms to conceal smashed data. Unlike fixed public transforms, these client-specific bases induce an exponentially large and effectively unsearchable function space, making brute-force inference by an adversary computationally infeasible.

- We provide a rigorous theoretical foundation for the design of these transforms. Specifically, we characterize the class of periodic functions that yield orthogonal, reversible, and numerically stable transformation matrices, and establish conditions (e.g., Chebyshev sampling) under which energy compaction and invertibility are preserved. This analysis ensures that obfuscation is both secure and utility-preserving, and cannot be trivially inverted, as detailed in Section 4.

- We conduct extensive empirical evaluation on both image and text benchmarks, demonstrating that SEAL withstands state-of-the-art reconstruction attacks while maintaining high task accuracy, as detailed in Section 5.

## 2 PRELIMINARIES AND RELATED WORK

### 2.1 SPLIT LEARNING

Split Learning (SL) is a distributed learning paradigm in which a deep neural network is partitioned into two segments: a *client-side* partial model and a *server-side* partial model. The client processes raw input $x$ through its portion of layers to produce an activation $h(x) = X$, called the *smashed data*, which is transmitted to the server. The server continues the forward pass, computes gradients, and sends back the gradient with respect to $X$ for client-side backpropagation. The server then continues training with the subsequent client. Compared to Federated Learning, clients in SL do not hold full models, which reduces client-side memory and compute requirements (Vepakomma et al., 2018; 2020; Thapa et al., 2021).

Variants of SL include *vanilla split learning*, *U-shaped split learning*, and *vertical split learning* (Vepakomma et al., 2018; Thapa et al., 2021; Hardy et al., 2017). In U-shaped SL, labels and the final layers of the neural network are kept on the client side so that even gradients arriving at the server do not directly reveal label information. While in vertical SL, the server also owns one initial partial model $h_s(x)$. These variants have been shown to be applicable in healthcare, federated inference settings, and cross-silo collaboration (Li et al., 2024; Thapa et al., 2022; Shan et al., 2021).

## 2.2 PRIVACY AND INFERENCE ATTACKS IN SPLIT LEARNING

Despite SL's design to avoid sharing raw inputs and labels, the shared smashed data and cut-layer gradients can still leak private information. Several attack classes have been studied. **Reconstruction or model inversion attacks** train inversion models or use optimization to recover client inputs from smashed activations (Pasquini et al., 2021; Xu et al., 2024; Luo et al., 2023; Zhu et al., 2023). For example, UNSPLIT shows that even an honest-but-curious server can reconstruct inputs and models with only the client architecture and smashed data (Erdoğan et al., 2022). In **Label inference attacks**, servers infer labels from activation or gradient patterns. Similarity-based attacks (Li et al., 2021) use cosine or Euclidean metrics, while in two-party and U-shaped SL, gradient responses can also leak labels (Liu et al., 2024). Finally, **active or hijacking attacks** let a malicious server manipulate activations or gradients to increase leakage (Pasquini et al., 2021). In this work, we defend against both passive and active inference attacks targeting input reconstruction.

## 2.3 PRIVACY DEFENSES IN SPLIT LEARNING

To protect the privacy of clients, multiple defenses have been proposed in the literature; we expand this list with possible candidates for obfuscation transform-based defenses, highlighting the possible shortcomings and limitations of these defenses.

Defenses such as NoPeek (Vepakomma et al., 2019) minimize dependence between inputs and smashed data, with decorrelation or mutual-information losses reducing correlations in activations. While effective against weaker adversaries, these defenses are circumvented by stronger reconstruction methods that exploit residual structure or side statistics (Pasquini et al., 2021; Xu et al., 2024).

Another class of defenses adds noise or applies differential privacy (DP). Gaussian or Laplace noise injected into activations or gradients provides DP guarantees, but noise accumulates across layers and rounds, severely reducing utility in deep models (Abadi et al., 2016). Even restricting DP to the cut layer often fails against modern inversion attacks (Luo et al., 2023).

We also consider the case of transform-based defenses, which map activations into known frequency bases such as DCT, PCA, or wavelets, often discarding high-frequency components. However, because these bases are fixed and public, the server can invert them (see Sect. 5.4), and their generalization across modalities and architectures is limited.

Finally, cryptographic techniques, including homomorphic encryption, MPC, and secret sharing, offer stronger leakage resistance but are computationally and communication-intensive, making them unsuitable for high-dimensional activations or resource-constrained clients (Vepakomma et al., 2018). Further, the server must process consistent inputs, so clients using different ciphertext keyspaces on their smashed data would impede the sequential nature of Split Learning training.

## 2.4 FREQUENCY TRANSFORMATIONS

A frequency transform represents a signal in an orthogonal basis, enabling operations such as compression, decorrelation, and noise filtering. The key idea is to project the signal into a new coordinate system where most of its energy is concentrated in a few coefficients, typically the low-frequency ones. This allows efficient representation while preserving the ability to reconstruct the original data. Formally, for a signal $O$ and orthogonal basis $B$, the transform is $S = B^\top \cdot O \cdot B$ with inverse $O = B \cdot S \cdot B^\top$, ensuring exact reconstruction.

Frequency transforms are widely used in signal processing and machine learning. They provide three essential properties: **orthogonality**, since the basis matrix $B$ satisfies $B^\top \cdot B = I$, ensuring independence of the transformed components in $S = B^\top \cdot O \cdot B$ and exact reversibility via $O = B \cdot S \cdot B^\top$; **energy compaction**, where most of the signal's total energy $\|O\|_2^2$ is concentrated in a small number of low-frequency coefficients of $S$, enabling efficient storage and transmission; and **decorrelation**, since the orthogonal projection reduces redundancy by making different entries of $S$ less correlated compared to the original data $O$. Together, these properties make frequency-domain representations more compact, robust, and informative than raw signals, with applications in image compression, audio analysis, and communication systems. Their analysis in SEAL is detailed in Section 4.

## 2.5 VALID PERIODIC FUNCTIONS

Not all functions can form valid transformation matrices. Non-periodic functions lack the regularity and orthogonality needed for clean decomposition, often leading to imbalance (non-vanishing integrals) and numerical instability. Hence, we restrict to periodic functions that guarantee transformations that are reversible, follow energy compaction, and are stable. Specifically, given function $f(x)$, it must satisfy two conditions: (i) **Integral Symmetry:** their integral over one period $T$ must vanish, $\int_0^T f(x)\,dx = 0$, ensuring balance and orthogonality and (ii) **Non-zero Chebyshev Nodes:** $f(x) \neq 0$ at all Chebyshev nodes $x = \frac{kT(2n+1)}{4N}$ for $k, n = 0, \ldots, N-1$, so that sampling on this non-uniform grid avoids linear dependencies and improves stability when constructing orthonormal bases.

## 3 THREAT MODEL

We consider a system with $N_c$ clients, each holding private datasets, unknown to other clients. A central server coordinates the training of DNN using the SL framework. SEAL integrates seamlessly with different SL paradigms (Vanilla Vepakomma et al. (2018) and U-shaped Thapa et al. (2021)), as it operates on the output of the client's input layers, regardless of subsequent framework operations. We define the adversary $\mathcal{A}$ as a compromised server attempting to reconstruct a client's private input data from exchanged intermediate representations.

We consider both the case of $N_c$ clients participating in the SL training (see Sect. 5.2) and for consistency with prior works Erdoğan et al. (2022); Xiao et al. (2017); Zhu et al. (2023); Li et al. (2024); Pasquini et al. (2021), we abstract the system into a two-party interaction between a victim client and the server. These attacks assume this abstraction because their protocol dynamics remain the same regardless of additional clients participating, and focus only on the interaction between a victim client and $\mathcal{A}$ (see Sect. 5.3). Under this threat model SEAL is applied locally with clients using their own private periodic function, never shared with others. Thus, privacy remains under client control.

The capabilities of $\mathcal{A}$ are: $\mathcal{A}$ is assumed to be fully knowledgeable: it can observe and store intermediate representations for reconstruction, has full knowledge of the DNN architecture (including client-side structure and parameter types, and server-side structure and parameter values), and may apply advanced reconstruction methods such as optimization, auxiliary datasets, or prior distributions. $\mathcal{A}$ is also aware of deployed defenses and can adapt its strategy, but it cannot tamper with benign client-side datasets or operations, relying solely on shared intermediate outputs.

## 4 SEAL

### 4.1 INTUITION AND HIGH-LEVEL DESIGN

SEAL is a client-side transformation defense that protects intermediate representations in SL from privacy attacks by the server. Before training, each client generates a private transformation matrix $Q$ from a chosen periodic function $f$ with period $T$. Construction involves (i) sampling $f$ at Chebyshev nodes to avoid redundancy, (ii) normalizing rows to equal scale, and (iii) applying Gram–Schmidt orthogonalization. The result is an orthonormal, energy-compacting, and numerically stable matrix that generalizes fixed transforms (e.g., DCT) but remains secret to the server. This one-time setup ensures efficiency during training.

During training and inference, the client applies $Q$ to its smashed data $X$, removes high-frequency coefficients to remove the irrelevant details leakage, and reconstructs the perturbed representation back into the feature space. This process is illustrated in Fig. 1.

### 4.2 PRE-TRAINING PHASE: CONSTRUCTING $Q$

This section elaborates on the core components of the pretraining phase in SEAL.

Figure 1: Overview of SEAL's workflow. The pre-training phase (Steps 1, 2, 3) initializes the transformation process by choosing $f$ with $T$ and calculating $N$; it outputs a transformation matrix $Q^{N \times N}$. The training phase (Steps 4, 5, 6, 7) integrates SEAL into the SL framework by operating on the client's smashed data and sending the perturbed smashed data $\tilde{X}$ to the server.

**Sampling with Chebyshev Nodes.** The first step in constructing the $Q$ is sampling the chosen periodic function $f$ at Chebyshev nodes rather than at uniformly spaced intervals. The reason is that Chebyshev nodes ensure the function satisfies the required conditions, i.e., **Integral Symmetry** and **Non-zero Chebyshev Nodes**. This non-uniform sampling avoids redundancy and numerical instabilities that can arise with uniform grids, especially when the periodic structure aligns poorly with the grid. Given a periodic function $f$ with period $T$ and smashed data of dimension $N \times N$, we construct a raw matrix $C \in \mathbb{R}^{N \times N}$:

$$C_{k,n} = f\left(\frac{k * T * (2n + 1)}{4N}\right), \quad k, n = 0, \ldots, N - 1$$

The resulting matrix $C^{N \times N}$ captures the periodic structure of $f$ while maintaining stability and symmetry, laying the foundation for building the transformation.

**Normalization.** Once $C^{N \times N}$ is generated, each row is normalized using the $L^2$-norm to enforce unit length. This ensures that all sampled basis vectors are on the same scale, preventing rows with larger magnitudes from dominating the transformation. By scaling each row to have a unit norm, we achieve geometric consistency and prepare the matrix for stable orthogonalization in the next step. Each row $C_{i,:}$ is normalized to unit magnitude:

$$\tilde{C}_{i,:} = \frac{C_{i,:}}{\|C_{i,:}\|_2}, \quad \|C_{i,:}\|_2 = \sqrt{\sum_j C_{i,j}^2}$$

**Orthonormalization via Gram–Schmidt.** Finally, the normalized vectors are processed through the Gram–Schmidt procedure to enforce orthogonality. Gram–Schmidt refines each vector by removing its projection onto previously computed ones, resulting in a set of orthonormal basis vectors. Next, we showcase the steps applied to $\tilde{C}$ to obtain an orthogonal matrix. For each column vector $u_k = \tilde{C}_{:,k}$, its projection onto previously processed column vectors $\{q_j\}_{j<k} = C_{:,j} \; \forall j < k$ is subtracted:

$$u_k \leftarrow u_k - \sum_{j=1}^{k-1} \frac{\langle q_j, u_k \rangle}{\langle q_j, q_j \rangle} q_j$$

and the result is normalized to unit length $q_k = u_k / \|u_k\|_2$. The resulting matrix $Q = [q_0, \ldots, q_{N-1}]$ satisfies $Q^\top \cdot Q = I$, guarantees orthogonality. Algorithm 1 integrates the above steps to construct a robust transformation matrix $Q^{N \times N}$ based on a chosen periodic function $f$.

---

**Algorithm 1:** Discrete Periodic Transform

---

**Input:** $f$ (periodic function), $T$ (period), $N$ (dimension of smashed data)
**Output:** $Q^{N \times N}$ (transform matrix)
$C \leftarrow \text{Chebyshev-Nodes}(f, T, N)$ ;            // Generate Chebyshev nodes
**for** $i \in [0, N-1]$ **do**
  $\tilde{C}_{i,:} \leftarrow \frac{C_{i,:}}{\|C_{i,:}\|_2}$ ;                       // Normalize rows
$Q \leftarrow \text{Gram-Schmidt}(\tilde{C})$ ;            // Orthonormalize columns
**return** $Q^{N \times N}$

---

---

**Algorithm 2:** SEAL: Perturbing Smashed Data

---

**Input:** $f$ (periodic), $T$ (period), $\omega$ (energy retention), client model $M$, input $I_{\text{data}}$
**Output:** $\tilde{X}$ (perturbed smashed data)
**Pre-training (one-time):**
$N \leftarrow \text{shape}(M.\text{output})$ ;                     // innermost dim.
$Q \leftarrow \text{DPT}(f, T, N)$ ;                              // Alg. 1
**Training / inference:**
$X \leftarrow M(I_{\text{data}})$ ;                       // compute smashed data
$Z \leftarrow Q \cdot X \cdot Q^\top$ ;                          // forward transform
$\mathcal{I} \leftarrow \text{retain\_indices}(Z, \omega)$ ;        // min set reaching $\omega$ energy
$Z_{i,j} \leftarrow 0 \; \forall (i,j) \notin \mathcal{I}$ ;            // mask HF coefficients
$\tilde{X} \leftarrow Q^\top \cdot Z \cdot Q$ ;                  // reconstruct to feature space
**return** $\tilde{X}$

---

Together, Chebyshev sampling, normalization, and orthonormalization form the core novelty of SEAL, letting each client build a private, well-conditioned transform that preserves utility while resisting inversion.

**A small note:** For notational simplicity, we present the smashed data as $N \times N$. However, SEAL can also generalize to non-square inputs: for $X \in \mathbb{R}^{\cdots \times H \times W}$, where $H$ and $W$ denote the least significant dimensions of the smashed data, clients construct $Q^1 \in \mathbb{R}^{H \times H}$ and $Q^2 \in \mathbb{R}^{W \times W}$, applying the separable transform $Z = Q^1 \cdot X \cdot Q^{2^\top}$ and $\tilde{X} = Q^{1^\top} \cdot Z \cdot Q^2$.

### 4.3 Training or Inference Phase: Perturbing Smashed Data

Algorithm 2 details the working of SEAL during the training (and inference) phase. SEAL operates as a lightweight client-side module after the cut layer. Given input data $I_{\text{data}}$, the client computes the smashed representation $X$ using its local model $M$. The precomputed DPT matrix $Q$ (from the pre-training phase) is then applied to transform $X$ into the frequency domain, producing $Z = Q \cdot X \cdot Q^\top$.

To protect privacy and retain utility, we leverage *energy compaction* to generate perturbed $\tilde{X}$. Most of the signal energy is concentrated in the low-frequency components, while high frequencies tend to capture fine-grained, identity-revealing details Lin et al. (2022); Wang et al. (2020). The total energy of the frequency matrix $Z$ is $E_{\text{total}} = \sum_{i=0}^{N-1} \sum_{j=0}^{N-1} |Z_{i,j}|^2$ and a client-chosen retention ratio $\omega$ controls how much of this energy to preserve. Following a zig-zag traversal of $Z$, we accumulate energy until the partial sum $E_{\text{low}}$ satisfies $E_{\text{low}}/E_{\text{total}} \geq \omega$, retaining only the most significant coefficients (for a detailed explanation of zig-zag traversal please see App. C and for a the exact and fast index search please see App. D and App. E). The remaining coefficients are zeroed out, producing a perturbed frequency matrix.

Finally, the inverse transform $\tilde{X} = Q^\top \cdot Z \cdot Q$ reconstructs a perturbed smashed representation, which is sent to the server in place of $X$. This process requires only matrix multiplications and coefficient masking, adding negligible overhead, while adaptively tuning the privacy–utility tradeoff via $\omega$.

Table 1: Baseline Evaluation of different Defenses, All values in percentage.

| | SEAL | NoPeekNN | DCOR | DPSGD |
|---|---|---|---|---|
| Scenario | MA | MA | MA | MA |
| no-defense | 94.6 | 94.6 | 94.6 | 94.6 |
| all-clients | 93.4 | 80.6 | 80.2 | 40.7 |
| 7-clients | 93.7 | 82.1 | 81.5 | 40.9 |
| 5-clients | 94.2 | 81.4 | 81.9 | 42.6 |
| 2-clients | 93.9 | 85.6 | 81.1 | 71.3 |
| 1-client | 94.1 | 85.7 | 83.8 | 72.2 |

## 5 EXPERIMENTS

This section evaluates SEAL against diverse reconstruction attacks, demonstrating its effectiveness in protecting client data during training and inference. We first outline the setup and quantitative metrics used to assert the effectiveness of SEAL (Sect. 5.1), we then analyze the base scenario of training a Transformer with no adversarial behavior, showcasing the minimal impact that our defense has on the overall SL framework (Sect. 5.2), followed by a quantitative and qualitative evaluation of SEAL against representative reconstruction attacks (Sect. 5.3). Lastly, we evaluate our defense against a knowledgeable adversary that uses known Discrete Transforms, or the exact secret function chosen by clients (Sect. 5.4).

### 5.1 EXPERIMENTAL SETUP

All experiments were conducted using the PyTorch deep learning library (Paszke et al., 2019). The simulated SL framework used a server equipped with four NVIDIA A6000 GPUs, an AMD EPYC 7773X CPU with 64 physical cores, and 768 GB of main memory. Clients were simulated using a Raspberry Pi 4 Model B, featuring a Broadcom BCM2711 quad-core Cortex-A72 processor at 1.5 GHz, and 8 GB of LPDDR4 RAM, with swap memory enabled. Communication of smashed data and gradients between machines was implemented using the Gloo[1] library.

**Metrics**: Our evaluation of SEAL leverages the following key metrics.

*Main Task Accuracy (MA)* measures the model's accuracy on the test dataset. It reflects the percentage of inputs for which the model delivers accurate predictions. This metric is essential to assess the utility impact of SEAL to the SL system.
*Structural Similarity (SSIM)* Wang et al. (2004) evaluates the perceived similarity between two images by focusing on structural information, which aligns more closely with human visual perception. It computes similarity based on three components: luminance, contrast, and structural correlation, combining them into a single score. The SSIM between the original input $x$ and the reconstructed sample $y$ is calculated as follows:

$$\text{SSIM}(x,y) = \frac{(2\mu_x\mu_y + C_1)(2\sigma_{xy} + C_2)}{(\mu_x^2 + \mu_y^2 + C_1)(\sigma_x^2 + \sigma_y^2 + C_2)}$$

Where $\mu_x$ and $\sigma_x^2$ are the mean and variance for $x$, similarly, $\mu_y$ and $\sigma_y^2$ are the mean and variance of $y$. The covariance between $x$ and $y$ is $\sigma_{xy}$; lastly, $C_1$ and $C_2$ are constants used to stabilize the calculations of luminance and contrast. The SSIM index ranges from -1 to 1, but in practice, values are usually between 0 and 1, where 1 indicates pixel-perfect reconstruction. The attacker aims to maximize SSIM, while SEAL strives to minimize it.

**Datasets.** Aligned and extending existing work on SL (Pasquini et al., 2021; Erdogan et al., 2022; Zhu et al., 2023; Xu et al., 2024), we leveraged five datasets (CIFAR-10, Tiny ImageNet, MNIST, FMNIST, and IBDM) to perform our experiments. For a detailed description of the datasets and model architectures used, please see App. A.

---

[1] https://github.com/pytorch/gloo

Table 2: Evaluation of different attacks in multiple dataset scenarios, All values in percentage.

| Attack | CIFAR-10 | | MNIST | | FMNIST | | ImageNet | |
|---|---|---|---|---|---|---|---|---|
| | MA | SSIM | MA | SSIM | MA | SSIM | MA | SSIM |
| FSHA | 70.1 | 9.3 | 82.3 | 14.8 | 83.5 | 13.1 | 49.2 | 8.1 |
| UnSplit | 76.8 | 10.9 | 96.7 | 13.5 | 91.2 | 12.5 | 57.3 | 12.2 |
| FSA | 76.8 | 11.2 | 97.3 | 20.3 | 89.4 | 17.9 | 62.3 | 16.3 |
| FORA | 76.5 | 10.1 | 97.3 | 10.2 | 88.6 | 13.4 | 62.4 | 10.9 |
| SDAR | 77.0 | 11.4 | 97.4 | 11.7 | 91.3 | 14.7 | 63.1 | 10.0 |

## 5.2 BASELINE EVALUATION

To demonstrate the broad applicability of our defense mechanism to more general SL scenarios, we extended our evaluation beyond image classification tasks, which typically rely on convolutional layers. Specifically, we assessed the use of our Discrete Periodic Transform (DPT) in a different domain, showcasing its compatibility with multidimensional signals (not only 2-D images). To achieve this, we simulate a real-world scenario where 10 clients trained a Transformer model (RoBERTa) on partitions of the IBDM dataset, with no adversarial actions performed by the server. We evaluated multiple configurations: (1) a standard SL setup without client-side perturbations or defense, (2) SL scenarios where one or more clients applied existing defenses and SEAL. Each client used a distinct $f$ when applying SEAL. As shown in Table 1, for SEAL the Main Task Accuracy (MA) was preserved across all cases, while other defenses were not able to operate properly, greatly reducing the utility of the model. The results confirm seamless integration of SEAL into SL without loss of utility.

## 5.3 ATTACKS EVALUATION

We consider five different reconstruction attacks: FSHA Pasquini et al. (2021), UnSplit Erdoğan et al. (2022), FSA Luo et al. (2023), FORA Xu et al. (2024) and SDAR Zhu et al. (2023). In our evaluation, we simulate 10 clients, each training on either CIFAR-10, MNIST, FMNIST, or Tiny ImageNet. The clients targeted consistently employ SEAL using a ratio of retained energy $\omega = 0.7$. Each experiment uses a different secret periodic function, showing the results are not tied to any specific choice of $f$. Table 2 shows that only on the MNIST and FMNIST datasets do attacks reach a higher average SSIM over the entire test-set, mainly due to their figures' predominantly black background boosting the score. Furthermore, Figure 2 qualitatively shows the reconstruction produced by the attacks on representative samples.

## 5.4 ADAPTIVE ATTACKS

Lastly, we evaluate SEAL under the presence of a knowledgeable adversary aware of the defense mechanism employed by the victim client. Two scenarios are considered: in the first, the client retains the secrecy of its periodic function, forcing $\mathcal{A}$ to guess the function used, and attempting to reverse the perturbation using known transforms (Discrete Cosine Transform). In the second, unrealistic scenario, $\mathcal{A}$ knows the periodic function and $\omega$ used by the victim client. Despite this advantage, we show that energy removal causes irreversible distortion to the signal, preventing the server from reconstructing perfect samples.

**Data Reconstruction Using DCT.** We extended FSHA Pasquini et al. (2021) by having the simulated client perform SEAL using the cosine function before passing its smashed data to the discriminator. For UnSplit Erdoğan et al. (2022), the clone model also employed the Discrete Cosine Transform (DCT) during the adaptive forward step. Additionally, FSA Luo et al. (2023) was modified, attempting to reverse the perturbation on the smashed data during the training phase of its autoencoder. FORA Xu et al. (2024) was updated by equipping the substitute client with SEAL. Lastly, SDAR Zhu et al. (2023) incorporated DCT in its simulator.

**Data Reconstruction With Full Knowledge.** We apply the same modification to all five evaluated attacks, as previously described, with the only difference being that SEAL in this scenario is implemented using the same secret function $f$ employed by the victim clients.

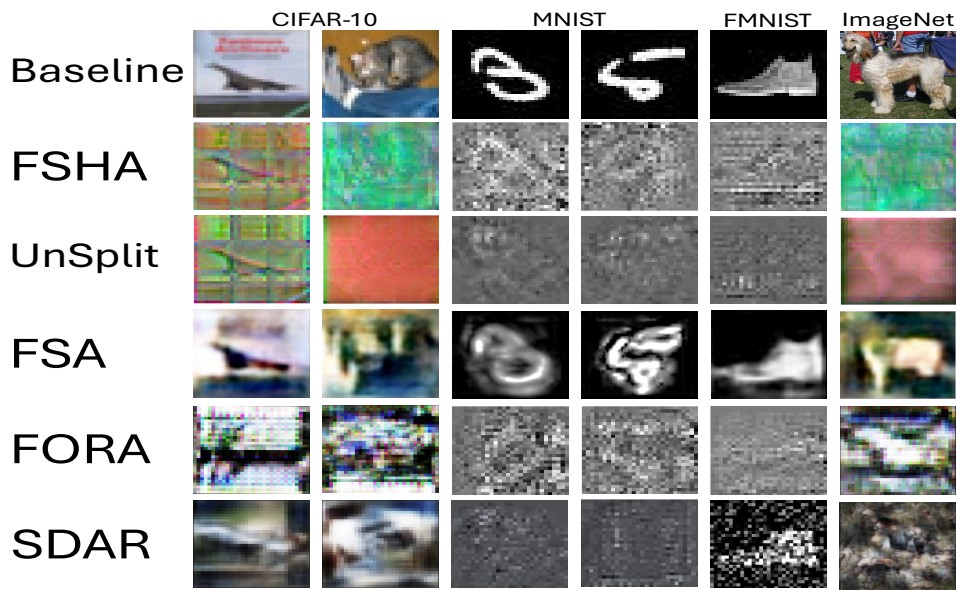

Figure 2: Baseline and reconstructed samples when SEAL is applied

Table 3: Adaptive attacks using DCT and the client's secret $f$, All values in percentage.

| Attack | | CIFAR-10 MA | CIFAR-10 SSIM | MNIST MA | MNIST SSIM | FMNIST MA | FMNIST SSIM | ImageNet MA | ImageNet SSIM |
|---|---|---|---|---|---|---|---|---|---|
| FSHA | DCT | 64.3 | 6.6 | 82.0 | 6.7 | 76.7 | 7.6 | 50.4 | 6.4 |
| | $f$ | 69.2 | 70.7 | 84.7 | 74.8 | 80.8 | 81.3 | 49.6 | 87.9 |
| UnSplit | DCT | 76.5 | 7.1 | 95.8 | 8.4 | 87.2 | 10.9 | 56.9 | 9.5 |
| | $f$ | 76.4 | 71.6 | 95.8 | 77.3 | 88.3 | 79.7 | 55.7 | 89.3 |
| FSA | DCT | 76.6 | 8.9 | 96.1 | 10.4 | 87.4 | 12.4 | 61.5 | 10.2 |
| | $f$ | 76.0 | 73.4 | 96.2 | 76.8 | 87.5 | 76.4 | 62.7 | 86.4 |
| FORA | DCT | 76.7 | 5.3 | 98.5 | 8.6 | 88.8 | 11.4 | 60.4 | 8.8 |
| | $f$ | 76.6 | 79.1 | 96.7 | 79.6 | 86.9 | 79.9 | 62.6 | 87.1 |
| SDAR | DCT | 76.6 | 6.4 | 97.3 | 7.9 | 89.1 | 10.8 | 60.0 | 5.9 |
| | $f$ | 75.7 | 73.0 | 97.5 | 81.9 | 87.4 | 80.1 | 62.1 | 88.5 |

**Adaptive Attack Result.** When $\mathcal{A}$ uses DCT, Table 3 shows a sharp SSIM drop with little impact on Main Task Accuracy (MA). In contrast, with knowledge of $f$, reconstruction improves substantially, with high-fidelity recovery on CIFAR-10 and Tiny ImageNet. It is important to note, however, that while these results may appear impressive, the practical feasibility of this attack is not achievable, since the function $f$ must remain secret to the clients and cannot be inferred from the smashed data (for a qualitative evaluation, please see Figure 3 in App. B).

## 6 CONCLUSION

In this work, we present SEAL, a client-side obfuscation framework for privacy in Split Learning by applying secret, client-specific periodic transforms to smashed data. These transforms create an exponential search space, making brute-force reconstruction of smashed data infeasible, while preserving orthogonality, invertibility, and stability. Experiments on image and text benchmarks show that SEAL resists state-of-the-art reconstruction attacks without sacrificing accuracy. We envision name as a foundation for privacy-preserving collaborative learning in edge intelligence, healthcare, and other privacy-critical applications, and as a new step toward bridging signal-processing theory with modern ML privacy guarantees.

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

Table 4: Client-Server Split of the DNN Architectures considered.

| | Number of Parameters | | |
| Model | Client | Server | Total |
|---|---|---|---|
| ResNet18 He et al. (2016) | 9 536 | 11 172 106 | 11 181 642 |
| ResNet50 Zagoruyko (2016) | 9 536 | 67 029 604 | 67 039 140 |
| RoBERTa Liu (2019) | 38 603 520 | 86 042 112 | 124 645 632 |

Xiaochen Zhu, Xinjian Luo, Yuncheng Wu, Yangfan Jiang, Xiaokui Xiao, and Beng Chin Ooi. Passive inference attacks on split learning via adversarial regularization. *arXiv preprint arXiv:2310.10483*, 2023.

## A  DATASETS AND MODEL ARCHITECTURES

### A.1  DATASET

We considered low-dimensional datasets, such as MNIST and FMNIST, as well as high-dimensional datasets, such as Tiny ImageNet, to analyze SEAL under different requirements to generalize our findings. Further, to extend outside of the simple image domain, we considered the IBDM dataset to tackle an NLP task in Split Learning. The datasets used are as follows:

*CIFAR-10* consists of 50 000 training and 10 000 test images of size $32 \times 32$ pixels, showing objects and animals belonging to 10 different classes Krizhevsky et al. (2009). As DNN, we use the widely adopted ResNet-50 architecture Zagoruyko (2016).

*Tiny ImageNet* consists of 100 000 training and 10 000 test images, spanning 200 different object categories Le & Yang (2015). As DNN, we use again use the ResNet-50 architecture Zagoruyko (2016).

*MNIST* consists of 60 000 training and 10 000 test grayscale images showing handwritten digits. As DNN, we use the widely adopted smaller ResNet-18 architecture He et al. (2016).

*FMNIST* is composed of 60 000 training and 10 000 test grayscale images, depicting various types of clothing across 10 classes Xiao et al. (2017). As DNN, we also use the widely adopted ResNet-18 architecture He et al. (2016).

*IBDM* Maas et al. (2011) is a dataset for binary sentiment classification, containing 25 000 training and 25 000 testing movie reviews texts. This dataset was used in conjunction with the model RoBERTa Liu (2019) on the sentiment analysis task.

### A.2  MODEL ARCHITECTURES AND CLIENT PARTITION

In our evaluations, we assess the performance of our defense across different model architectures: ResNet18, ResNet50, and RoBERTa. As Table 4 shows, we choose to partition the client-side computation at the earliest possible feature extraction layer, e.g., the first residual block for ResNet architectures, and after the word embeddings generation for RoBERTa. This decision is two-fold: first, it more closely represents the limited computational capabilities of the clients, and second, it maximizes the advantage for the proposed adversaries Pasquini et al. (2021); Erdoğan et al. (2022); Xu et al. (2024); Luo et al. (2023); Zhu et al. (2023). By cutting the computation at these early layers, we increase the client's vulnerability, facilitating more effective reconstructions in no-defense scenarios. This strategy enables us to rigorously evaluate the robustness of our defense on realistic client-side constraints.

## B  QUALITATIVE EVALUATION OF ADAPTIVE ATTACKS

Figure 3 showcases how $\mathcal{A}$ employing an incorrect secret function to simulate SEAL, leads to worse reconstructions, while in the case where $f$ is known to $\mathcal{A}$, it is able to obtain meaningful reconstructions. This further highlights the need for clients employing SEAL, to keep their $f$ secret.

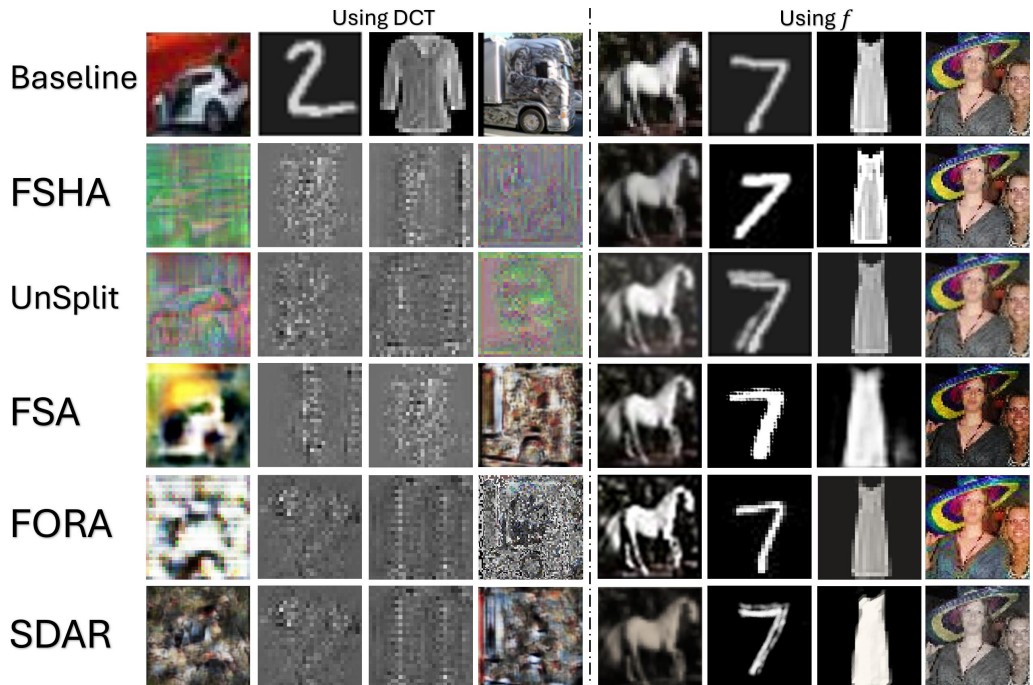

Figure 3: Adaptive attacks employing DCT (left) and the client's secret $f$ (right)

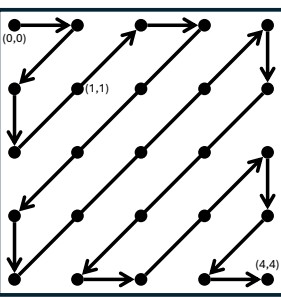

Figure 4: "Zig-Zag" Reading of a Frequency Matrix

## C    TRAVERSE OF A FREQUENCY MATRIX

Below is defined the algorithm used to traverse a Frequency Matrix, following Figure 4. Algorithm 3 takes as input the previous position in the matrix and the desired direction and returns the new position in the reading. This algorithm is used by both Algorithms 4 and 5 to explore a frequency matrix and calculate the indices of its low-frequency components.

---

**Algorithm 3:** ZigZag Traverse

---

**Input:** $i, j$ (Current indices), $d$ (Current direction), $N$ (Square matrix dimension)
**Output:** $i, j, d$ (New indices and direction)
$r, c \leftarrow i + d, j - d$ ;        // Compute next position based on direction
**if** $0 \leq r < N$ **and** $0 \leq c < N$ **then**
  | $i, j \leftarrow r, c$ ;          // Update indices if within matrix bounds
**else**
  | **if** $d = -1$ **then**
  |   | Moving "up-right";
  |   | **if** $j = N - 1$ **then**
  |   |   | Hit the right boundary;
  |   |   | $i \leftarrow i + 1$ ;                                          // Move down
  |   | **else**
  |   |   | Hit the top boundary;
  |   |   | $j \leftarrow j + 1$ ;                                          // Move right
  | **else**
  |   | Moving "down-left";
  |   | **if** $i = N - 1$ **then**
  |   |   | Hit the bottom boundary;
  |   |   | $j \leftarrow j + 1$ ;                                          // Move right
  |   | **else**
  |   |   | Hit the left boundary;
  |   |   | $i \leftarrow i + 1$ ;                                          // Move down
  | $d \leftarrow -d$ ;                              // Change direction
**return** $i, j, d$

---

## D   ENERGY COMPACTION ALGORITHM DURING TRAINING

As follows is an outline of Algorithm 4, which produces the set of indexes of low-frequency components of the frequency matrix $Z$ that retain the ratio of energy $\omega$. The algorithm uses the function *ZigZag Traverse*$(i, j, d, N)$ defined in Algorithm 3 to navigate $Z$ following the structure defined in Figure 4.

---

**Algorithm 4:** Energy Compaction Property

---

**Input:** $Z$ (Frequency matrix), $\omega$ (Desired energy retention ratio)
**Output:** $Indexes$ (Set of matrix indices to retain)
$N \leftarrow \text{shape}(Z)$ $E_{tot} \leftarrow \sum_{i=0}^{N-1} \sum_{j=0}^{N-1} |Z_{i,j}|^2$ ;     // Total energy of transform matrix
$E_{low} \leftarrow 0$ ;                              // Initialize retained energy
$Indexes \leftarrow \{\}$ ;                              // Initialize retained indices
$i, j \leftarrow 0, 0$ ;                                // Start at first position
$d \leftarrow -1$ ;      // Traversal direction:  $-1$ = up-right, $1$ = down-left
**while** $\frac{E_{low}}{E_{tot}} < \omega$ **do**
  | $E_{low} \leftarrow E_{low} + |Z_{i,j}|^2$ ;                // Add energy of current element
  | $Indexes \leftarrow Indexes \cup \{(i,j)\}$ ;                    // Retain current index
  | $(i, j, d) \leftarrow$ **ZigZag Traverse**$(i, j, d, N)$ ;          // Compute next position
**return** $Indexes$

---

# E ENERGY COMPACTION ALGORITHM BEFORE TRAINING

To reduce computational overhead, it is possible to precompute the set of low-frequency indices (indexes) before the Split Learning training begins, using the Discrete Periodic Transform (DPT) matrix and a randomly generated signal $R$. These indices can then be reused in every iteration during training. This approach leverages the empirical stability of low-frequency components when the transform matrix $DPT$ remains fixed, significantly accelerating computations on resource-constrained client devices. However, as stated, this optimization is empirical and comes at the cost of losing the security guarantee provided by dynamically determining at each iteration which components to keep using the $\omega$ ratio.

---

**Algorithm 5:** Alternative Energy Compaction Property

---

**Input:** $DPT$ (Transform matrix), $\omega$ (Desired energy retention ratio)
**Output:** $Indexes$ (Set of matrix indices to retain)
$N \leftarrow \text{shape}(DPT)$ Initialize random matrix $R^{N \times N}$ $Z \leftarrow DPT^T \cdot R \cdot DPT$ ;
  // Transform of the random matrix
$E_{tot} \leftarrow \sum_{i=0}^{N-1} \sum_{j=0}^{N-1} |Z_{i,j}|^2$ ;       // Total energy of transform matrix
$E_{low} \leftarrow 0$ ;                // Initialize retained energy
$Indexes \leftarrow \{\}$ ;           // Initialize retained indices
$i, j \leftarrow 0, 0$ ;               // Start at first position
$d \leftarrow -1$ ;    // Traversal direction: $-1$ = up-right, $1$ = down-left
**while** $\frac{E_{low}}{E_{tot}} < \omega$ **do**
    $E_{low} \leftarrow E_{low} + |Z_{i,j}|^2$ ;       // Add energy of current element
    $Indexes \leftarrow Indexes \cup \{(i,j)\}$ ;       // Retain current index
    $(i, j, d) \leftarrow \textbf{ZigZag Traverse}(i, j, d, N)$ ;      // Compute next position
**return** $Indexes$

---

# F ABLATION STUDY OF $\omega$

## F.1 IMPACT OF $\omega$ ON PRIVACY AND UTILITY

In prior experiments, we varied the periodic function $f(x)$ while fixing $\omega$ at 0.7 to balance privacy and utility. In Table 5, we instead fix $f(x)$ and examine how varying $\omega$ affects SEAL on the CIFAR-10 and MNIST datasets and in Table 6 on the FMNIST and Tiny ImageNet datasets. As $\omega$ increases, more high-frequency components are retained, resulting in higher Main Task Accuracy (MA) but also improved reconstruction quality for the adversary $\mathcal{A}$ (SSIM). At $\omega = 1$, SEAL is effectively disabled, offering no protection (No Defense). Importantly, as detailed in Section 2.4, the number of frequency components removed does not scale linearly with $\omega$. A 0.1 decrease in $\omega$ corresponds to the removal of significantly more than 10% of the elements in the frequency matrix $Z$. Consequently, even at $\omega = 0.9$, where MA remains unaffected, we observe a noticeable drop in SSIM, indicating reduced reconstruction fidelity. Conversely, values of $\omega$ below 0.7, such as 0.4 and 0.1, produce an alleged increase in privacy, but at the cost of degraded MA. This is due to the removal of essential low-frequency components, which leads to impaired learning. Nevertheless, MA does not collapse entirely, since the server and other clients do not deviate in their behavior for training, except in the case of a fully malicious server, such as in FSHA Pasquini et al. (2021), where the server actively alters gradients during training.

810
811
812
813
814
815
816
817
818
819
820
821
822
823
824
825
826
827
828
829

Table 5: Efficacy of SEAL for different values of $\omega$ on CIFAR-10 and MNIST datasets, All values in percentage.

| Attack | $\omega$ Dataset | 1.0 MA | SSIM | 0.9 MA | SSIM | 0.4 MA | SSIM | 0.1 MA | SSIM |
|---|---|---|---|---|---|---|---|---|---|
| FSHA | CIFAR-10 | 71.2 | 90.3 | 71.4 | 50.2 | 35.9 | 6.2 | 33.3 | 5.4 |
| | MNIST | 86.5 | 98.2 | 86.7 | 57.7 | 59.3 | 8.2 | 54.6 | 7.5 |
| UnSplit | CIFAR-10 | 77.0 | 92.5 | 77.0 | 45.8 | 49.6 | 7.3 | 45.8 | 6.4 |
| | MNIST | 99.0 | 97.1 | 97.4 | 50.9 | 65.3 | 7.9 | 60.0 | 6.6 |
| FSA | CIFAR-10 | 77.1 | 94.6 | 76.9 | 53.4 | 50.2 | 7.1 | 46.1 | 5.8 |
| | MNIST | 98.7 | 98.3 | 97.5 | 58.0 | 65.6 | 8.5 | 60.1 | 6.7 |
| FORA | CIFAR-10 | 77.2 | 93.6 | 77.0 | 51.6 | 50.3 | 8.9 | 45.7 | 5.8 |
| | MNIST | 99.1 | 98.2 | 97.6 | 56.5 | 65.8 | 7.2 | 60.1 | 6.2 |
| SDAR | CIFAR-10 | 77.0 | 92.7 | 77.1 | 51.1 | 49.9 | 5.4 | 46.4 | 5.1 |
| | MNIST | 98.9 | 98.1 | 97.8 | 55.4 | 65.1 | 7.9 | 59.8 | 7.6 |

830
831
832
833
834
835
836
837
838
839
840
841
842
843
844
845
846

Table 6: Efficacy of SEAL for different values of $\omega$ on FMNIST and Tiny ImageNet datasets, All values in percentage.

| Attack | $\omega$ Dataset | 1.0 MA | SSIM | 0.9 MA | SSIM | 0.4 MA | SSIM | 0.1 MA | SSIM |
|---|---|---|---|---|---|---|---|---|---|
| FSHA | FMNIST | 83.6 | 96.5 | 82.7 | 56.0 | 45.2 | 6.3 | 39.9 | 6.1 |
| | ImageNet | 59.7 | 91.0 | 57.2 | 40.1 | 37.8 | 7.9 | 30.3 | 6.5 |
| UnSplit | FMNIST | 91.2 | 97.2 | 91.3 | 58.5 | 51.3 | 8.5 | 47.8 | 8.3 |
| | ImageNet | 65.4 | 95.4 | 65.3 | 45.2 | 40.4 | 9.6 | 39.4 | 7.0 |
| FSA | FMNIST | 91.1 | 98.2 | 91.0 | 60.4 | 50.8 | 9.1 | 47.8 | 8.7 |
| | ImageNet | 65.4 | 94.3 | 65.4 | 43.3 | 41.4 | 9.2 | 38.7 | 7.3 |
| FORA | FMNIST | 91.4 | 97.8 | 90.9 | 57.2 | 50.7 | 7.4 | 47.3 | 6.8 |
| | ImageNet | 65.3 | 95.7 | 65.2 | 45.1 | 40.9 | 8.7 | 38.7 | 8.4 |
| SDAR | FMNIST | 91.2 | 97.9 | 91.1 | 59.9 | 51.0 | 8.3 | 47.6 | 7.9 |
| | ImageNet | 65.2 | 93.6 | 65.3 | 45.8 | 40.8 | 8.4 | 40.0 | 5.2 |
