# OpenReview forum: "Split, Not Spilled: Practical Obfuscation-Based Privacy-Preserving Split Learning"
_ICLR.cc/2026/Conference — ICLR 2026 Conference Withdrawn Submission_

### Official Review · Reviewer_u9ym · 2025-10-22

**Soundness:** 1
**Presentation:** 1
**Contribution:** 2
**Rating:** 2
**Confidence:** 5

**Summary:**

This paper proposes a new defense for split learning systems that transforms the smashed data into a frequency matrix and retains only a subset of important components for reconstruction, thereby reducing information leakage.

**Strengths:**

1. Proposes a method to reduce information leakage by truncating the frequency matrix.

2. Evaluates multiple attack methods.

**Weaknesses:**

However, I still have some concerns about the proposed method, and the evaluation is too weak to be convincing.

1. The proposed method can be summarized as transforming the smashed data into the frequency domain, retaining the most important portion ($w$), and then reconstructing the matrix. How does this approach compare to simpler methods, such as DCT or DST transformations, when applying the same portion ($w$) retention and reconstruction? The paper should evaluate their accuracy and defense performance against DPT.

2. In the evaluation presented in Section 5.2, no hyperparameters are specified for the three defenses, and these parameters directly influence the privacy-utility trade-off. Therefore, simply comparing accuracy is not an adequate evaluation. The proposed defense should be compared with other methods in terms of both accuracy and attack performance under varying hyperparameter settings. Additionally, more defenses should be included in the evaluation, such as random noise methods [1,2] and Ressfl [3].

3. In the evaluation in Section 5.3, no attack results are reported for the no-defense baseline, so the poor attack outcomes lack a proper comparison standard and may simply reflect the 10-client setup being unfavorable to attackers. The authors should also evaluate a single-client scenario—consistent with prior work (UnSplit, FSHA, FORA, SDAR)—since it is more advantageous to attackers and would provide a clearer baseline.

4. The evaluated attacks are primarily feature-space-based (e.g., FSHA, FORA) and may be heavily influenced by the proposed defense. Therefore, PCAT [1], which relies solely on the server-side model, should also be included in the evaluation.

5. Regarding the adaptive attacks, the evaluated DCT-based one is nearly identical to the standard attacker, as the attacker may not use $w$. Therefore, three additional scenarios should be considered: (a) DCT with random $w$, (b) DCT with exactly the same $w$, and (c) the proposed transformation $f$ with random $w$. Including these scenarios with the full-knowledge attacker case, would yield a more comprehensive evaluation of the defense.

6. In Appendix A.2, it is stated that only shallow layer partitions are evaluated. More partition layers should be tested, as the defense’s privacy-utility trade-off may vary across different depths.

7. Minor issues: Figure 2 is distorted, and the font sizes are inconsistent.

[1] Gao, Xinben, and Lan Zhang. "{PCAT}: Functionality and data stealing from split learning by {Pseudo-Client} attack." 32nd USENIX Security Symposium (USENIX Security 23). 2023.

[2] Xu, Xiaoyang, et al. "A stealthy wrongdoer: Feature-oriented reconstruction attack against split learning." Proceedings of the IEEE/CVF conference on computer vision and pattern recognition. 2024.

[3] Li, Jingtao, et al. "Ressfl: A resistance transfer framework for defending model inversion attack in split federated learning." Proceedings of the IEEE/CVF conference on computer vision and pattern recognition. 2022.

**Questions:**

1. Why does SL under FSHA attack still show accuracy (MA) in Tables 2 and 3? FSHA hijacks the feature space, so the server does not use any classification loss, and no server model is trained according to the paper and its code.

2. In the Adaptive Attacks, does the DCT group use $w$ to control the preserved frequency matrix?

3. In Section 5.1, does the attacker use auxiliary data drawn from the same distribution or from a different distribution?
How are the private and auxiliary datasets partitioned for the experiments?

4. Why are the UnSplit results in Figure 3 so good? In the original paper, the reconstructed colors appear worse.

---

### Official Review · Reviewer_vB3D · 2025-10-29

**Soundness:** 1
**Presentation:** 2
**Contribution:** 1
**Rating:** 2
**Confidence:** 5

**Summary:**

The paper proposed a new frequency domain transformation based on obfuscation functions to defend against inversion attacks in split learning. Abundant experiments were evaluated to show the effectiveness of the solution.

**Strengths:**

1. Authors studied an important question regarding privacy-preserving split learning.
2. Authors conducted a lot of experiments for the evaluation

**Weaknesses:**

1. The idea of using an unknown obfuscation function in split learning is not new. See [1], [2]. Solutions are pretty effective but they are not compared in the paper.
2.Baselines mentioned in 2.3 omit recent work on defense in split learning. [1-5]

[1] Xu, Hengyuan, Liyao Xiang, Hangyu Ye, Dixi Yao, Pengzhi Chu, and Baochun Li. "Permutation equivariance of transformers and its applications." In Proceedings of the IEEE/CVF Conference on Computer Vision and Pattern Recognition, pp. 5987-5996. 2024.
[2] Oh, Seungeun, Sihun Baek, Jihong Park, Hyelin Nam, Praneeth Vepakomma, Ramesh Raskar, Mehdi Bennis, and Seong-Lyun Kim. "Privacy-preserving split learning with vision transformers using patch-wise random and noisy cutmix." arXiv preprint arXiv:2408.01040 (2024).
[3] Le, Trung, Hao Fang, Jingyuan Li, Tung Nguyen, Lu Mi, Amy Orsborn, Uygar Sümbül, and Eli Shlizerman. "Spint: Spatial permutation-invariant neural transformer for consistent intracortical motor decoding." arXiv preprint arXiv:2507.08402 (2025).
[4] Lee, Wonjun, Bumsub Ham, and Suhyun Kim. "Maximizing the Position Embedding for Vision Transformers with Global Average Pooling." In Proceedings of the AAAI Conference on Artificial Intelligence, vol. 39, no. 17, pp. 18154-18162. 2025.
[5] Yao, Dixi, and Baochun Li. "Is Split Learning Privacy-Preserving for Fine-Tuning Large Language Models?." IEEE Transactions on Big Data (2024).

**Questions:**

1. What does "passive adversaries" mean?
2. Based on the threat model described in section 3, it is the same as the previous assumption. Why do authors argue that previous defenses assumed different adversaries?
3. With the transform function, features are transformed to the frequency domain. But the model weights are still in the spatial domain. How do you do the training/inference and maintain the performance?
4. What models do you use? Which layer do you cut the model in? What are the detailed architectures of the models you used?

---

### Official Review · Reviewer_cufs · 2025-10-31

**Soundness:** 2
**Presentation:** 2
**Contribution:** 2
**Rating:** 2
**Confidence:** 3

**Summary:**

This paper proposes to use a matrix transformation to hide information in the intermediate representations to preserve the privacy of client data in the setting of split learning. In particular, the paper constructs an orthogonal matrix from a periodic function. Then, the intermediate representations are mapped via this matrix, where certain entries are dropped to preserve privacy, and then mapped back using the inverse transformation. Experiments show that this method is effective in defending against existing attacks while achieving better utility than existing defense methods.

**Strengths:**

1. Interesting problem statement.
2. Using a theory-motivation transformation to hide information.
3. Seems to achieve good defense performance without hurting utility.
4. Experiments cover a wide range of defenses and attacks.

**Weaknesses:**

1. "Obfuscation" seems a strong word for what this paper is doing.
1. No formal security guarantee.
2. Delivery could be improved, especially the figures. They are not in the correct size or aspect ratio.
3. Insufficient discussion on the choice of parameters.

Please see the questions sections for more details.

**Questions:**

1. Sec 2.2: "In this work, we defend against both passive and active inference attacks targeting input reconstruction.". Sec 3: "A is also aware of deployed defenses and can adapt its strategy, but it cannot tamper with benign client-side datasets or operations, relying solely on shared intermediate outputs." This is a self-contradiction.
2. I'm skeptical of the security guarantees. There is no formal security guarantee provided in the paper. Since Q is orthogonal, i.e., $Q^\top Q=I$, then without the masking in $Q X Q^\top$, we would have perfect reconstruction from $Q^\top Z Q=X$. Therefore, the only information loss to protect privacy is from the dropping of the non-significant coefficient. Then, for this map from $X$ to $\tilde X$ to be "invertible", does this claim mainly come from the fact that we are masking the matrix?
3. I'm not entirely sure in what ways the properties of $Q$ are important for privacy and utility. Imagine I do PCA on the intermediate representations and drop dimensions with low eigenvalues, and then convert them back. Does that also achieve a similar goal? In fact, since each time the X is different, this approach also have changing transformations all the time. Is it important that $Q$ is not data dependent?
4. What is the parameter $\omega$ chosen? It seems to be the most important parameter as it controls how much information we drop, and the paper does not discuss or justify its choice. I am also curious about its impact on utility and defense effectiveness.

---

### Official Review · Reviewer_eTNA · 2025-10-31

**Soundness:** 3
**Presentation:** 1
**Contribution:** 2
**Rating:** 2
**Confidence:** 4

**Summary:**

- This paper presents an algorithm for split-learning with privacy
- The main idea is that each client uses an independent and secret discrete periodic transforms, parameterized by a client-specific function f
- The transform is applied after the client-side layers, and in the transformed space, high-frequencies are removed, before transforming back to the original space, and sending it to the server
- The intuition is that low frequency modes are necessary for learning, and high frequency ones are less necessary, but leak more image-specific information
- Empirical results show good performance when evaluated again different attacks

**Strengths:**

- Method description is well written, idea is clear and well motivated
- The proposed defense is cheap (requiring only client-side matmuls)
- Empirical defense quality against several attacks appears good
- Results on textual data in addition to 2d data show that SEAL could be used in different modalities

**Weaknesses:**

- Undescriptive table captions. For example table 1: "Baseline Evaluation of different Defenses, All values in percentage." What task is being shown here? This table is pointed to in section 5.2, which deals with textual data, but the table caption is not descriptive. Another example is table 2: "Evaluation of different attacks in multiple dataset scenarios, All values in percentage." This caption should be more specific.

- Missing baseline accuracy results. Section 5.1 describes several vision datasets, but results for these are only give for SEAL. Where is the accuracy results for these baselines?

- Missing baseline defense results. In order to evaluate the tradeoff between accuracy and reconstruction quality of SEAL, we need to compare the defense/accuracy of baseline defenses. In particular, it would be useful to include a plot with "reconstruction SSIM" on one axis, and "main task accuracy" on the other, and plot SEAL along with baseline defenses on that plot (at various privacy hyperparameter choices). Without attacks performed on baselines, we cannot evaluate the quality of SEAL.

- Missing description of choice of f. How are these random periodic functions chosen?

Generally speaking, it seems that this paper may have been rushed or these baselines results were purposefully omitted. I think the idea of SEAL is compelling, however the evaluation of SEAL compared to baselines is below what is necessary for acceptance in its current state. I think with proper comparisons (and good results), this paper can be accepted.

**Questions:**

- Is it correct that during inference, you also using SEAL? I.e. do inference users also have randomized transforms? What is the effect of not using SEAL during inference and only during training?

---

### Note · Authors · 2025-11-21

I have read and agree with the venue's withdrawal policy on behalf of myself and my co-authors.